# Attitude sustains longer than subjective norm and perceived behavioral control: Results of breast cancer screening educational intervention

**Rojana Dhakal**[1]*, **Chiranjivi Adhikari**[2,3], **Prabha Karki**[1], **Nirmala Neupane**[1], **Pooja Bhandari**[1], **Aditi Gurung**[1], **Nisha Shrestha**[4], **Nandaram Gahatraj**[2], **Niranjan Shrestha**[2], **Niranjan Koirala**[5], **Govind Subedi**[6]

1 Department of Nursing, School of Health and Allied Sciences, Pokhara University, Pokhara, Gandaki Province, Nepal, 2 Department of Public Health, School of Health and Allied Sciences, Pokhara University, Pokhara, Gandaki Province, Nepal, 3 Department of Public Health, Indian Institute of Public Health Gandhinagar (IIPHG), Gandhinagar, Gujarat, India, 4 Department of Nursing, Pokhara Nursing Campus, Institute of Medicine, Tribhuvan University, Pokhara, Gandaki Province, Nepal, 5 Department of Natural Products Research, Gandaki Province Academy of Science and Technology, Pokhara, Nepal, 6 Central Department of Population Studies, Tribhuvan University, Kritipur, Bagmati Province, Nepal

* rojanadkl@pu.edu.np, rojanabuddhi2@gmail.com

**Data Availability Statement:** All relevant data are within the paper and its Supporting Information files.

## Abstract

Breast malignancy is the most frequent carcinoma among females across the world and third-most in Nepal. Early diagnosis of breast cancer through breast health awareness and self-examination, in addition to mammography screening, is a highly feasible and useful technique in poorly resourced settings. However, their intentions, whether to modify behaviors or actions, remain debatable and less explained in the literature. So, we aimed to assess how long an educational intervention affects women's intention to do a breast self-examination (BSE) and mammography screening. After assessing feasibility, one ward was assigned to the intervention (IG; ward number 30) and control group (CG; ward number 33), and then with inclusion criteria, a total of 360 females (180 each in IG and CG) aged 40–75 years enrolled in the study. After the baseline assessment, participants in the IG were delivered an hour-long breast cancer screening-related lecture-discussion- demonstration session that included BSE and mammography, aided with a silicone dummy. The session was carried out by the female trained nurses. Outcome data were obtained at the baseline, 4, 8, and 12 months following the intervention. Attitudes, perceived behavioral controls (PBCs) and behavioral intents (BIs) of both mammography and BSE at baseline were similar in both IG and CG except in case of subjective norms (SNs). Intents of BSE remained effective for 4 months, whereas for mammography, it was effective only at 4 and 12 months. Moreover, attitudes toward both tests remained intact for 4, 8, and 12 months (p = < .05) consistently. With regards to PBCs, women having good control remained only for 4 months in both screening tests. Further, regarding SN, significant mean changes were observed at 4 and 12 months in BSE, and only at 4 months in mammography screening. The session was effective in sustaining BSE and mammography intentions for at least 4 months. To retain

**Funding:** We received financial support (FRG number: 73/74-HS-15) from the University Grants Commission, Nepal to carry out the study. Pokhara University's School of Health and Allied Sciences disbursed the funds directly to the principal investigator's official account, which was given by the University Grants Commission directly to the school. The silicone breast model was purchased with funds provided by the University Grants Commission. The funding and other institutions had no role in the study design, data collection and analysis, the decision to publish, or the preparation of the manuscript."

**Competing interests:** The authors have declared that no competing interests exist.

the effects longer (up to 12 months), additional educational strategies focusing on subjective norms and perceived behavioral controls of both tests are highly warranted.

## Introduction

Globally, breast tumor is the topmost malignancy in females [1]. Breast cancer is the third most common cancer in Nepal, with an incidence rate of 9.6% among total cancer cases and a mortality rate of 7.7% in the year 2020 [2].

Early screening of breast cancer is an effective strategy for early identification of the tumor, which contributes to early treatment and enhances survivorship. Breast self-examination (BSE), clinical breast exam (CBE), and mammogram x-ray are some of the most commonly used screening tests. These tests aids in the early detection of cases that are still curable [3]. BSE can begin as early as the age of 20 to examine and feel one's breast for any abnormalities [4, 5]. Women should begin regular screening starting at the age of 45 years old or the women aged between 40 and 44 years could do breast cancer screening yearly if they wish to. Women aged 45–54 years old should be screened annually, and women aged 55 and older should be tested every two years with mammography screening and should continue as long as their overall well-being is satisfactory or they have a life expectancy of ten years or more [6, 7].

A Chinese study found that cultural beliefs play an important role in screening for breast cancer. Out of 494 Chinese women only, 27.5% of women performed BSE, 36.4% CBE, 23.5% mammography, and 40% performed ultrasonography [8]. Research conducted in Butwal Sub-metropolitan city, only 31% had ever heard of BSE and 19.2% had ever practiced it. This demonstrates a significant gap in BSE knowledge and practice among Nepalese women [9]. According to a study in Nepal, there is a lack of awareness about the symptoms, early diagnosis, and screening tests for breast cancer [10]. Thus, they need the education to improve their knowledge and screening behavior.

The theory of planned behavior (TPB) is one of several health behavior theories that have been developed to predict health behavior [11]. BSE was predicted solely by intention [12]. Attitude, subjective norms, and perceived behavioral controls are the three components of this theory [13]. A study done in Kathmandu, Nepal, found that out of 500 women, 3.4%, 7.2% and 14.4% of women had undergone mammography, CBE, and BSE respectively. If a woman had favorable attitudes, high subjective norms, and greater control of behavior, she was more likely to undergo screening measures [14]. Furthermore, according to Iranian research, Jeihooni and colleagues discovered that an educational intervention based on TPB increased the intention to get mammography screening for breast cancer [15]. Furthermore, a study from Western Iran discovered that 13% of 408 women had mammography done once. Control over behavior and behavioral intention were predictors of mammogram uptake [16]. Early identification of cancer cases and raising awareness about the need for screening procedures minimize the number of people who go to the hospital late for screening and treatment [17]. Intent to perform breast cancer screening in Nepal showed the women had poor screening behavior [14]. In Cypriot women, age, education, perceived control over behavior, and self-efficacy are some factors that determine the intent to perform mammography [18]. Many studies conducted in the USA revealed that poor knowledge of breast cancer screening, lack of trust in hospitals and doctors, language barriers, transportation issues, embarrassment, and low income are the factors that act as a barrier to utilizing mammography screening [19]. Further, a study from Tennessee showed that lack of awareness of screening methods (58%), not knowing of a place to go for screening (59%), and fear of being diagnosed with cancer (23%), are the main barriers

to having mammography [20]. Thus, there is a need for educational intervention to improve breast cancer screening knowledge and practice and reduce cancer-related mortality. To date, there has been very little study on the effectiveness of educational intervention in determining women's desire to go for breast cancer screening in Nepal. Hence, our study's main aim is to find out the effect of educational intervention on the intention to engage in breast cancer screening and how long intervention affects women's intent to do BSE and mammography based on a theory of planned behavior.

# Methods and materials

## Study design and setting

A quasi-experimental study was conducted among women aged 40–75 years in Pokhara, Nepal. First, the community wards were randomly assigned to the intervention and control groups, and then respondents who meet the inclusion criteria were selected by using purposive sampling. The sample size was determined using the OpenEpi sample size calculator for a cross-sectional, cohort, and clinical trials study with a power of 80%, and a confidence level of 95%, with the assumption of a study of mammography utilization [21], with a percentage of unexposed and exposed with outcomes of 27% and 43%, respectively. The calculated sample size was 154 in the intervention group and 154 in the control group. The total calculated minimum sample size was 308. Considering the 16% dropout rate, the final targeted sample size was 360.

## Inclusion and exclusion criteria

The following were the inclusion criteria: women aged between 40 and 75 years, not having a history of benign or malignant breast disease, residents of Pokhara Metropolitan 30 and 33 for the past 6 months, speaking Nepali, and not participated in any educational session training related to breast cancer. Women who refused to participate in the study, did not know Nepali, had a history of mental illness, and refused to continue participating in the follow-up study were all exclusion criteria. The respondents of the intervention and control groups were matched by age.

## Data collection instruments

We adapted the questionnaire from literature based on constructs of the theory of planned behavior; had considerable reviews, received expert's comments, and also formative research has been carried out among 40 women from ward number 10 of Pokhara, which was not included in the study. An oncologist, psychiatric nurse, gynecologist, nurse midwife, and health promotion specialist made up the expert panel. The questionnaire was prepared in the English language and then translated into the Nepali language and back-translated to English. Based on the literature, intention scales include a total of 16 items for BSE (attitude; 6 items, subjective norms; 6 items and Perceived behavioral control; 4 items) and there were18 items (attitude; 8 items, subjective norms; 6 items and perceived behavioral control; 4 items) for mammography screening. Each construct had calculated Cronbach's alpha values greater than 0.7. Furthermore, the designed educational set was validated by experts, which comprises both theoretical and practical lessons. A five-point Likert scale was used to measure the intention of doing BSE and mammography. On a scale of one to five, one represents "strongly disagree", and five represents "strongly agree" [12, 22–26].

## Exposure variables and assessment

**Attitude, perceived behavioral control, subjective norms, and behavioral intention.**
Icek Ajzen (1985, 1991) created the Theory of Planned Behavior (TPB) as a comprehensive paradigm for understanding, predicting, and explaining behaviors. Behavioral intentions are instantaneously decided by behavioral intent, and the three elements that influence behavioral intentions are attitude, subjective norms, and perceived behavioral control [23, 25, 26]. An attitude toward behavior consists of behavioral beliefs and outcome behavior; subjective norms of behavior consist of normative beliefs and motivation to comply; while perceived behavior control includes control belief strengths and power of control factors. Each construct was multiplied by its sub-constructs independently, and the overall score of the main constructs was computed. After that, the total score for the behavioral intention was determined by adding all three key constructs: attitude, subjective norms, and perceived behavior control, and the final score for the behavioral intention was calculated. A score less than or equal to the median value was classified as unfavorable attitudes, while a score above than the median value was considered as favorable attitudes. A similar scoring system was applied to the subjective norms, perceived behavior control, and behavioral intention (Table 1).

## Outcome variables

The intention to do monthly breast self-examination and go for mammography every two years was the primary result of this research.

**Potential confounders.** *Knowledge of breast cancer*. Knowledge was measured using a 51-item breast cancer awareness measure. Responses were measured using the nominal scale of "true", "false" and "do not know". The overall knowledge score was calculated by summing the number of correct responses with possible scores ranging from 0 to 51. Respondents were given one (1) point for each correct answer and zero (0) for each wrong or unsure response.

*Socio-demographic and reproductive characteristics of participants*. Variables like age, religion, marital status, ethnicity, educational status, type of family, and occupation were included in the demographic factors. Reproductive history includes age at menarche, age at first childbirth, menopause, the history of breast cancer in oneself and one's family, etc.

*Breast self-examination proficiency skills*. The proficiency of BSE was measured through an observation checklist that consisted of both techniques of inspection and palpation of practicing BSE. Breast self-examination proficiency was calculated by summing the number of correct

**Table 1. Cut-off score and categorization of variables.**

| SN | BSE categorization | Mammography Categorization |
|---|---|---|
| 1. | Attitude | Attitude |
| | Unfavorable ($\leq$ 48) | Unfavorable ($\leq$ 40) |
| | Favorable (> 48) | Favorable (> 40) |
| 2. | Subjective norms | Subjective norms |
| | Non-compatible with referents ($\leq$ 32) | Non-compatible with referents ($\leq$ 31) |
| | Compatible with referents (>32) | Compatible with referents (>31) |
| 3. | Perceived Behavioral Control | Perceived Behavioral Control |
| | Poor control ($\leq$ 23) | Poor control ($\leq$ 32) |
| | Good Control (> 23) | Good Control (> 32) |
| 4. | Behavioral Intention | Behavioral Intention |
| | Low intention ($\leq$ 102) | Low intention ($\leq$ 101) |
| | High intention (>102) | High intention (>101) |

responses, with possible values ranging from 0 to 38. Each completely correct response obtained a score of two points, each incomplete but correct response was worth one point, and the incorrect or unperformed task obtained a score of zero.

*Breast cancer screening behavior*. The women were asked if they performed BSE regularly. Similarly, during the period of study, whether, she underwent for mammography screening.

### Data collection

Data was collected through face-to-face interviews at baseline, 4 months, 8 months, and 12 months in both the intervention and control groups. The educational intervention consists of a one-hour session based on TPB. The intervention group received a repeated intervention at baseline, 4, and 8 month time frame. Breast anatomy, facts on breast cancer, signs and symptoms, risk factors, screening measures, treatment, preventive measures, BSE, and Mammography were all discussed throughout the educational session. Breast self-examination technique was taught and demonstrated by a trained female nurse. The intervention group was educated using a variety of methods, including lecture, discussion, and demonstration. The American Cancer Society guidelines [6, 27], WHO [28–30], and International Agency for Research on Cancer [31] recommendations were used to create the lecture and demonstration guide. Participants demonstrated the BSE techniques (explain and show) and palpation (demonstrate on a silicon model). The illiterate women were educated through discussion, showing pictures and demonstrating the BSE palpation in a model, and showing standing position.

### Ethics statement

The Institutional Review Committee (IRC number 28-75-76) of Pokhara University granted ethical approval. Formal permission from respected wards of the Pokhara Metropolitan was obtained. The participants gave oral and written consent. The comparison group received the same informative session at the end of the research. Privacy and confidentiality were maintained. Women participated voluntarily in the study, and their names were not used in the study reports. Also, they were informed about the publication of the study results.

### Data analysis

Epi-data Version 3.1 was used to enter the data, which was exported to SPSS version 25 for analysis. Descriptive and inferential statistics were calculated based on the nature of the variables and objectives of the study. The efficacy of the intervention was assessed using independent and paired t-tests. For statistical significance, a p-value was set at $< 0.05$.

### Results

Table 2 summarizes the socio-demographic characteristics of the women included in the study. The average age in the intervention and control groups was 49 years. Hinduism was practiced by the vast majority of women. In the intervention and control groups, 25.6% and 30.6% of individuals were illiterate, respectively. Chi-square test confirmed that there were no significant association between the two groups in any of the demographic variables ($p = \geq 0.05$)

Table 3 demonstrates that 6.7% of women in the intervention group and 2.8% of women in the control group expressed favorable attitudes towards performing BSE at the start of the study. In the experimental group's subsequent posttests at 4, 8, and 12 months, favorable attitudes towards conducting BSE were 21.1%, 18.9%, and 22.8%, respectively, following the intervention. In addition, some changes were found in the control group, with 16.1% of individuals intending to conduct BSE at 12 months of the pretest. After the educational intervention, the

**Table 2. Demographic variables of respondents (n = 360).**

| Characteristics | Intervention group | Control group | p-value |
|---|---|---|---|
| | Frequency (percent) | Frequency (percent) | |
| **Age (years)** | | | |
| ≤ 50 years | 110 (61.1%) | 115 (63.9%) | 0.58 |
| >50 years | 70 (38.9%) | 65 (36.1%) | |
| Mean ± SD | 49.4±9.3 | 49.6±8.1 | |
| **Religion** | | | |
| Hindu | 162 (90.0%) | 167 (92.8%) | 0.49 |
| Buddhist | 16 (8.9%) | 13 (7.2%) | |
| Muslim | 1 (0.6%) | 0 (0.0%) | |
| Christian | 1 (0.6%) | 0 (0.0%) | |
| **Marital Status** | | | |
| Married | 168 (93.3%) | 170 (94.4%) | 0.82 |
| Divorced/widow* | 12 (6.7%) | 10 (5.6%) | |
| **Ethnicity** | | | |
| Brahmin | 98 (54.4%) | 101 (56.1%) | 0.14 |
| Chhetri | 18 (10.0%) | 19 (10.6%) | |
| Dalit | 12 (6.7%) | 23 (12.8%) | |
| Muslim | 1 (0.6%) | 0 (0.0%) | |
| Janjati | 51 (28.3%) | 37 (20.6%) | |
| **Education** | | | |
| Illiterate | 46 (25.6%) | 55 (30.6%) | 0.35 |
| Can read or write | 53 (29.4%) | 57 (31.7%) | |
| Literate | 81 (45.0%) | 68 (37.8%) | |
| **If Literate** | | | |
| SEE and below | 62 (76.5%) | 62 (91.2%) | .05 |
| Higher Secondary | 16 (19.8%) | 5 (7.4%) | |
| Bachelor and above | 3 (3.7%) | 1 (1.5%) | |
| **Type of family** | | | |
| Nuclear | 87 (48.3%) | 91 (50.6%) | 0.43 |
| Joint | 86 (47.8%) | 86 (47.8%) | |
| Extended | 7 (3.9%) | 3 (1.7%) | |

intervention group's compatibility with referents improved in posttest 1 (4 months) and posttest 2 (8 months). However, the control group showed unexpected improvements, compatibility with referents increasing to 78.9% at posttest3 from 35.6% at baseline. Surprisingly, both groups had higher levels of perceived behavioral control and behavioral intention to execute BSE.

Table 4 shows that 47.8% of women from intervention groups had a favorable attitude toward performing mammography screening in the pretest. After the intervention, the experimental group had a higher positive score of 77.2% at 4 months, 84.4% at 8 months, and 83.0% at 12 months. In contrast, the control group participants also had increased positive attitudes toward performing mammography, but had less favorable scores than the intervention group, 43.3% from baseline to 68.3% in the 12 months. Next, with regards to subjective norms in the pretest, 55.0% of intervention participants and 44.4% of control group participants were compatible with referents. The participants in both groups had increased compatibility with the appropriate reference group for their intention to perform mammography.

**Table 3. Descriptive statistics to perform breast self-examination based on the constructs of the theory of planned behavior at different endpoints (n = 360).**

| Categories | Intervention group | | | | Control group | | | |
|---|---|---|---|---|---|---|---|---|
| | Baseline | 4 month | 8 month | 12 month | Baseline | 4 month | 8 month | 12 month |
| *Attitude towards behavior* | | | | | | | | |
| Unfavorable (≤48) | 168 (93.3%) | 142 (78.9%) | 146 (81.1%) | 132 (77.2%) | 175 (97.2%) | 174 (96.7%) | 153 (85%) | 151 (83.9%) |
| Favorable (>48) | 12 (6.7%) | 38 (21.1%) | 34 (18.9%) | 39 (22.8%) | 05 (2.8%) | 06 (3.3%) | 27 (15.0%) | 29 (16.1%) |
| *Subjective Norm (SN)* | | | | | | | | |
| Non-Compatible (≤32) | 80 (44.4%) | 57 (31.7%) | 53 (29.4%) | 74 (43.3%) | 116 (64.4%) | 119 (66.1%) | 38 (21.1%) | 38 (21.1%) |
| Compatible (>32) | 100 (55.6%) | 123 (68.3%) | 127 (70.6%) | 97 (56.7%) | 64 (35.6%) | 61 (33.9%) | 142 (78.9%) | 142 (78.9%) |
| *Perceived Behavioral Control (PBC)* | | | | | | | | |
| Poor (≤23) | 80 (44.4%) | 7 (3.9%) | 6 (3.3) | 10 (5.8%) | 100 (55.6%) | 101 (56.1%) | 11 (6.1%) | 11 (6.1%) |
| Good (>23) | 100 (55.6%) | 173 (96.1%) | 174 (96.7%) | 161 (94.2%) | 80 (44.4%) | 79 (43.9%) | 169 (93.9%) | 169 (93.9%) |
| *Behavioral Intention (BI)* | | | | | | | | |
| Low (≤102) | 81 (45.0%) | 30 (16.7%) | 23 (12.8%) | 46 (26.9%) | 101 (56.1%) | 112 (62.2%) | 33 (18.3%) | 35 (19.4%) |
| High (>102) | 99 (55.0%) | 150 (83.3%) | 157 (87.2%) | 125 (73.1%) | 79 (43.9%) | 68 (37.8%) | 147 (81.7%) | 145 (80.6%) |

Regarding perceived behavioral control, 91.7% and 93.3% of the intervention and control groups, respectively had poor control over getting screened for breast cancer by mammography. Furthermore, after the first intervention, perceived control in the intervention group participants was slightly increased; 21.7% had good perceived control, but 80% of participants had poor perceived control in subsequent posttest 3 to get screened through mammography. Majority of 90% of participants in the control group had poor perceived control in posttest3 of getting screened by mammography. With regards to behavioral intention, 54.4% and 44.4% of the intervention and control groups, respectively, had a high intention of performing mammography in the future in the pretest. Following the intervention, (87.2%) of women in post-test 1; (37.2%) in post-test 2; and (83.6%) in post-test 3 had a high intention to get screened by mammogram x-ray.

Baseline mean scores of attitude, perceived behavioral control, and behavioral intention of mammography were almost similar (p = >.05), however regarding BSE, all of those were similar (p = >.05) except in subjective norms (p = 0.02). After educational intervention, significant differences were observed in the attitude (at 4, 8 and 12 months), subjective norms, and

**Table 4. Comparative frequency and percentage distribution of TPB constructs at different endpoints of mammography intention (n = 360).**

| Categories | Intervention group | | | | Control group | | | |
|---|---|---|---|---|---|---|---|---|
| | Baseline | 4 month | 8 month | 12 month | Baseline | 4 month | 8 month | 12 month |
| *Attitude towards behavior* | | | | | | | | |
| Unfavorable (≤40) | 94 (52.2%) | 41 (22.8%) | 28 (15.6%) | 29 (17.0%) | 102 (56.7%) | 139 (77.2%) | 53 (29.4%) | 57 (31.7%) |
| Favorable (>40) | 86 (47.8%) | 139 (77.2%) | 152 (84.4%) | 142 (83.0%) | 78 (43.3%) | 41 (22.8%) | 127 (70.6%) | 123 (68.3%) |
| *Subjective Norm (SN)* | | | | | | | | |
| Non-Compatible (≤31) | 81 (45.0%) | 50 (27.8%) | 54 (30.0%) | 61 (35.7%) | 100 (55.6%) | 90 (50.0%) | 45 (25.0%) | 41 (22.8%) |
| Compatible (>31) | 99 (55.0%) | 130 (72.2%) | 126 (70.0%) | 110 (64.8%) | 80 (44.4%) | 90 (50.0%) | 135 (75.0%) | 139 (77.2%) |
| *Perceived Behavioral Control (PBC)* | | | | | | | | |
| Poor (≤32) | 165 (91.7%) | 141 (78.3%) | 161 (89.4%) | 137 (80.1%) | 168 (93.3%) | 169 (93.9%) | 160 (88.9%) | 162 (90.0%) |
| Good (>32) | 15 (8.3%) | 39 (21.7%) | 19 (10.6%) | 34 (19.9%) | 12 (6.7%) | 11 (6.1%) | 20 (11.1%) | 18 (10.0%) |
| *Behavioral Intention (BI)* | | | | | | | | |
| Low (≤101) | 82 (45.6%) | 23 (12.8%) | 113 (62.8%) | **28** (16.4%) | 100 (55.6%) | 81 (45.0%) | 142 (78.9%) | 75 (41.7%) |
| High (>101) | 98 (54.4%) | 157 (87.2%) | 67 (37.2%) | 143 (83.6%) | 80 (44.4%) | 99 (55.0%) | 38 (21.1%) | 105 (58.3%) |

**Table 5. Comparison of mean scores of TPB constructs in mammography and breast-self-examination at different end-points (n = 360).**

| Variables | Time | Mammography | | | Breast self-examination | | |
|---|---|---|---|---|---|---|---|
| | | Intervention | Control | p value | Intervention | Control | p value |
| Attitude | Baseline | 42.85(7.73) | 41.93(6.15) | .217 | 44.17 (9.75) | 43.65(8.60) | 0.59 |
| | 4 month | 49.20 (9.76) | 41.36 (6.56) | .001* | 50.37(9.30) | 46.98 (7.01) | .001* |
| | 8 month | 51.73 (11.62) | 44.06 (6.01) | .001* | 50.24(8.50) | 45.75 (7.99) | .001* |
| | 12 month | 57.39 (11.16) | 43.93 (6.48) | .001* | 52.16 (10.30) | 45.78 (8.0) | .001* |
| Subjective Norms | Baseline | 33.19(11.75) | 32.36(10.80) | .48 | 34.14 (12.06) | 31.42(10.31) | 0.02* |
| | 4 month | 38.26 (13.04) | 34.07 (11.82) | .002* | 38.25(12.32) | 31.78 (11.57) | .001* |
| | 8 month | 37.58 (14.30) | 35.53 (6.59) | .082 | 38.81 (11.98) | 38.26 (8.32) | .0613 |
| | 12 month | 35.88 (13.58) | 35.84 (6.75) | .969 | 34.82 (14.30) | 38.22 (8.38) | .007* |
| Perceived Behavioral Control | Baseline | 27.73(8.59) | 27.57(6.15) | 0.83 | 23.41(9.49) | 22.54(7.54) | 0.338 |
| | 4 month | 34.03 (6.98) | 31.28 (5.47) | .001* | 32.85 (6.77) | 20.54 (5.03) | .001* |
| | 8 month | 25.03 (4.32) | 24.43 (4.02) | .178 | 31.70 (6.45) | 32.12 (5.80) | 0.509 |
| | 12 month | 25.90 (5.18) | 24.38 (3.90) | .002* | 31.69 (6.93) | 32.08 (5.76) | 0.569 |
| Behavioral Intention | Baseline | 103.77(20.46) | 101.87(13.96) | 0.30 | 101.73(24.96) | 97.62(19.73) | 0.08 |
| | 4 month | 121.50 (23.87) | 106.72 (17.28) | .001* | 121.48 (21.99) | 99.32 (16.60) | 0.001* |
| | 8 month | 97.12 (17.37) | 92.04 (13.79) | .002* | 120.76 (21.17) | 116.15 (17.25) | 0.024* |
| | 12 month | 119.19 (21.68) | 104.16 (11.38) | .001* | 118.69 (24.13) | 116.09 (17.40) | 0.247 |

*significance, changes are calculated based on baseline value minus follow-up time points, #n = 171 in the intervention group at 12 months, p values are derived from independent samples t-test

behavioral intention to perform BSE at 4 and 8 months respectively (p = <0.05). Furthermore, no significant changes were observed in the perceived control of behavior after the intervention at 8 and 12 months of posttests (Table 5).

With regards to mammography, the independent t-test revealed that there were no significant differences observed between the intervention and control groups in all the constructs of TPB at the baseline (p>0.05). The changes in the average scores in attitudes (p = .001 at 12 months) and behavioral intention (p = .001 at 12 months) were found in all the independent pairs. With regards to subjective norms, after the first intervention, there were statistically significant differences observed (p = 0.002 at 4 months), and participants were compatible with referents to perform mammography. Next, there was a significant enhancement in the mean score of perceived behavior control after the first (p = .001) and third interventions (p = .002) (Table 5).

Table 6 represents the mean changes in the intention to perform breast self-examination from baseline between groups, after intervention following at 4, 8, and 12 months. Within the control group attitude, subjective norms, perceived behavioral control, and intent to perform BSE all improved significantly from baseline to different endpoints. The mean changes in the attitude constructs were found significant at 4 months (p = .015), at 8 months (p = .001), and at 12 months (p = .001). However, in the case of subjective norms, there were changes significantly observed in the 4 months (p = .017) and 12 months (p = .001) endpoints. Similarly, regards to the capacity to perform BSE and the intent to perform BSE, the changes were observed only at 4 months endpoints (p = .001 and p = .001, respectively).

Table 7 shows the changes in the intention to perform mammography from baseline between groups, after intervention at 4, 8, and 12 months. From baseline to three different endpoints, the intervention group's attitude towards undergoing mammography all improved significantly (p = .001). Further, significant changes were observed in subjective norms at 4months (p = .018), perceived behavioral control at 4 months (p = .010) and behavioral intent to undergo mammogram were observed in the 4 months and 12 months (p = .001).

**Table 6. Mean changes in constructs of theory of planned behavior intent to perform breast self-examination by time and group (n = 360).**

| Variables | Time | Intervention | Control | Difference (95% CI) | p value |
|---|---|---|---|---|---|
| Attitude | 4 month | -6.20 (13.36) | -3.33 (8.16) | -2.86 (-5.16, -.57) | .015* |
| | 8 month | -6.06 (11.90) | -2.10(11.06) | - 3.96 (-6.34, -1.58) | .001* |
| | 12 month | -8.26 (14.82) | -2.13(10.93) | -6.13 (-8.86, -3.41) | .001* |
| Subjective Norms | 4 month | -4.11(17.74) | -.36(10.96) | -3.74(-6.80,- .68) | .017* |
| | 8 month | -4.67(16.07) | -6.84 (12.32) | 2.17 (-.79, 5.14) | .151 |
| | 12 month | -1.15(17.86) | -6.80 (12.52) | 5.64(2.41, 8.87) | .001* |
| Perceived Behavioral Control | 4 month | -9.43 (11.81) | 2.0 (5.55) | - 11.43(-13.35,-9.52) | .001* |
| | 8 month | -8.28 (10.49) | -9.58(9.27) | -1.29 (-.75, 3.34) | .216 |
| | 12 month | -8.50 (11.20) | -9.53(9.25) | -1.03 (-1.11, 3.18) | .345 |
| Behavioral Intention | 4 month | -19.75(33.20) | -1.70 (16.91) | -18.05(-23.51,-12.58) | .001* |
| | 8 month | -19.02(29.32) | -18.52(23.58) | .50 (-6.01, 5.01) | .859 |
| | 12 month | -17.92(34.92) | -18.47(23.53) | .54(-5.67, 6.77) | .863 |

*significance, changes are calculated based on baseline value minus follow-up time points, #n = 171 in intervention group at 12 months, p values are derived from independent samples t-test

## Discussion

Behavioral intentions to carry out both tests in the future were found to be significantly different (p < .05). The significant changes in BI toward having mammography were found at 4 and 12 months, which might be due to maturation and diffusion of treatment. The findings of the present study are supported by the study of Ali Khani Jeihooni et al. [15], which shows women's intent to undergo mammography was increased 6 months after the intervention.

Intent to do BSE in the present study showed that significant changes were only observed at 4 months. The findings were supported by a study on breast cancer in Zahedan, which showed that after educational intervention, mean scores of knowledge, attitude, perceived behavior control, behavioral intention, and adopting screening behaviors improved in the intervention group compared to the control group [32].

**Table 7. Mean changes in constructs of theory of planned behavior intent to perform mammography by time and group (n = 360).**

| Variables | Time | Intervention | Control | Difference (95% CI) | p value |
|---|---|---|---|---|---|
| Attitude | 4 month | -6.35 (11.49) | -.57 (8.29) | -6.92 (-9.00, -4.84) | .001* |
| | 8 month | -8.88 (14.63) | -2.12 (8.68) | - 6.76 (-9.26, - 4.27) | .001* |
| | 12 month | -14.69 (13.90) | -1.99(8.98) | -12.69 (-15.14, -10.25) | .001* |
| Subjective Norms | 4 month | -5.06 (16.68) | -1.71(8.83) | -3.35 (-6.12,- .58) | .018* |
| | 8 month | -4.38(17.33) | -3.16 (12.30) | -1.22 (-4.33, 1.89) | .441 |
| | 12 month | -2.95(17.56) | -3.47(12.65) | .51 (-2.68, 3.72) | .750 |
| Perceived Behavioral Control | 4 month | -6.30 (11.32) | -3.71 (7.23) | - 2.58 (-4.55,-.61) | .010* |
| | 8 month | 2.70 (9.37) | 3.13 (7.75) | -.43 (-2.21, 1.35) | .633 |
| | 12 month | 1.60 (10.68) | -3.18(7.42) | -1.57 (-3.49, .34) | .108 |
| Behavioral Intention | 4 month | -17.72(28.77) | - 4.85 (13.93) | -12.87(-17.55,-8.18) | .001* |
| | 8 month | -6.65 (24.47) | - 9.83(17.88) | -3.17 (-7.62, 1.26) | .161 |
| | 12 month | -16.04(30.03) | -2.28(16.68) | -13.75 (-18.82, - 8.68) | .001* |

*significance, changes are calculated based on baseline value minus follow up time points, n = 171 in the intervention group at 12 months, p values are derived from independent samples t- test

Interestingly, attitudes remained consistently significant in both mammography and BSE at the 4, 8, and 12-month periods. The attitude towards a behavior is more individual. It differs according to how beneficial, pleasurable, or pleasant they believe the behavior is [26]. People develop favorable or positive mindsets as an outcome of educational instruction. This finding is similar to prior research, suggesting that intervention work enhances the attitudes scores, as demonstrated by Ali Khani Jeihooni et al. [15], showed that the intervention groups had a significant increase in attitude toward performing mammography screening, also, Steadman et al. [33] indicating the change towards favorable attitudes. Further, a favorable attitude was achieved in the intervention group by describing the circumstances of breast cancer, asking and answering questions, and having conversations to identify the beliefs and behavior outcomes to modify the negative attitudes towards positive attitudes [15].

In terms of mean scores of subjective norms, a significant change was observed only at 4 months for mammography, whereas for BSE, significant changes in mean scores were found at 4 and 12 months. According to a study carried out in Latina Washington State recommendation from family, friends, and spouses increase intention and uses of mammography [34], and subjective norms worked as a motivational factor by following important people as valued in one's life. In the current study, the intervention group's compatibility with reference was observed in each subsequent follow-up at 4, 8, and 12 months for both test, whereas similar changes occurred in the control group almost comparable with the study by Sargazi [32] where subjective norms did not change. In the present study, spouses, family members, and children are the most significant peoples who have a role in increasing compatible behavior, similar to a study in China [35] and another study from Isfahan Iran [36].

The education session of the present study was not held for spouses, children, and family members but discussed how participants had been influenced by their peers and their drive to comply with their advice [13]. The findings of the present study corroborate the study by Yamile et al. [34], in which the importance of subjective norms in performing breast cancer screening was highlighted. In addition, a study by Mason and White [13] supports the idea that subjective norms played an important role in the prediction of BSE behavior, which is similar to the present study.

The term "perceived behavioral control" (PBC) refers to a person's views about their ability to engage in a certain activity. It was stated that when people believe they have control over their conduct, they get motivated and do such activities even in difficult conditions [23, 37]. Earlier research by Norman and Hoyle [12], perceived behavior control is an important factor of TPB constructs in predicting the BSE as well as mammography intentions [38]. Based on our findings educating and increasing women's abilities to conduct BSE will greatly enhance their behavior intention. In the current study, lecturing and discussing the issue, showing pictures, and conducting BSE in a silicone model improved the intervention group's perceived behavior control, and attitudes [38–41]. In present study, PBC remained sustained for 4 months only, for both tests.

Previous research on mammography screening and perceived control in Iranian women found that while mammography screening is not entirely under one's control, the factors such as embarrassment, fear of breast cancer diagnosis, family support, radiation exposure, painful technique, cost, and long waits in public mammography centers work as psychological barriers to accessibility [42, 43]. The findings of a study on the intention to maintain mammography adherence showed that women who have stronger sense of controls are more likely to obtain mammogram [44]. A study of Iranian women regarding mammography screening [15], a study from Malaysia related to breast cancer screeni [ng [40], and a study of women 40 years of age and older in Isfahan, Iran [45], all supported the improvement in perceived control following an educational intervention.

The strength of this study is that we tested TPB constructs with direct measurement of intention in BSE and mammography. However, the study is limited to women between the ages of 40–75 years, and the random assignment of participants was not done. Moreover, the reason for not performing BSE and mammography screening, or cost analysis for mammography could not be assessed.

## Conclusion

Overall, the results of the study showed that adopting an educational intervention based on the theory of planned behavior was effective in sustaining BSE and mammography intentions for only 4 months. However, the attitude towards performing both tests remained effective for up to a year. To retain the effects longer (up to 12 months or longer), additional educational strategies focusing on subjective norms and perceived behavioral controls of both tests are highly warranted. It is suggested to use this theory to improve behaviors regarding breast cancer screening. Also, attitude and perceived behavioral control are the main constructs that determine behavior, however it is recommended to carry out research on individual constructs of TPB.

## Supporting information

**S1 File.**
(SAV)

## Acknowledgments

We would like to express our deepest gratitude to the study participants, the University Grants Commission of Nepal, and the 6th and 7th batches of B.Sc. Nursing students at the School of Health and Allied Sciences, Pokhara University, for their enormous support during the research periods.

## Author Contributions

**Conceptualization:** Rojana Dhakal, Chiranjivi Adhikari, Prabha Karki.

**Formal analysis:** Rojana Dhakal, Chiranjivi Adhikari, Nandaram Gahatraj, Niranjan Shrestha.

**Funding acquisition:** Rojana Dhakal, Nirmala Neupane, Pooja Bhandari, Aditi Gurung, Nisha Shrestha.

**Investigation:** Rojana Dhakal, Prabha Karki, Nirmala Neupane.

**Methodology:** Rojana Dhakal, Chiranjivi Adhikari, Govind Subedi.

**Project administration:** Rojana Dhakal.

**Resources:** Prabha Karki, Nirmala Neupane, Pooja Bhandari, Aditi Gurung, Nisha Shrestha.

**Software:** Rojana Dhakal, Nandaram Gahatraj, Niranjan Shrestha.

**Supervision:** Niranjan Koirala, Govind Subedi.

**Writing – original draft:** Rojana Dhakal.

**Writing – review & editing:** Rojana Dhakal, Chiranjivi Adhikari, Prabha Karki, Nirmala Neupane, Pooja Bhandari, Aditi Gurung, Nisha Shrestha, Nandaram Gahatraj, Niranjan Shrestha, Niranjan Koirala, Govind Subedi.

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
