## [Decision Letter · Decision Letter 0]

20 Jun 2022

PONE-D-22-05192The effect of educational intervention on intention to perform breast self-examination and mammography among women of Pokhara Nepal: application of theory of planned behaviorPLOS ONE

Dear Dr. Rojana,

Thank you for submitting your manuscript to PLOS ONE. After careful consideration, we feel that it has merit but does not fully meet PLOS ONE’s publication criteria as it currently stands. Therefore, we invite you to submit a revised version of the manuscript that addresses the points raised during the review process.

 Please submit your revised manuscript by **31 July 2022**. If you will need more time than this to complete your revisions, please reply to this message or contact the journal office at plosone@plos.org. Please include the following items when submitting your revised manuscript:A rebuttal letter that responds to each point raised by the academic editor and reviewer(s). You should upload this letter as a separate file labeled 'Response to Reviewers'.A marked-up copy of your manuscript that highlights changes made to the original version. You should upload this as a separate file labeled 'Revised Manuscript with Track Changes'.An unmarked version of your revised paper without tracked changes. You should upload this as a separate file labeled 'Manuscript'.

We look forward to receiving your revised manuscript.

Kind regards,

Siew Chin Ong, PhD

Academic Editor

PLOS ONE

**Journal requirements:**

2. PLOS ONE does not copy edit accepted manuscripts (https://journals.plos.org/plosone/s/criteria-for-publication#loc-5). To that effect, please ensure that your submission is free of typos and grammatical errors.

“We would like to express our gratitude to the University Grants Commission (UGC-Nepal) of Nepal for awarding faculty research grants (FRG-73/74-HS-15) to carry out this research. In addition to this, we would like to express the deepest gratitude for study participants and B.Sc. Nursing 6th and 7th batch students of School of Health and Allied Sciences, Pokhara University for enormous support during the phase of data collection and intervention. RD, NN, PB, AG, NS received the grants to carry out the research from UGC-Nepal.”

“This study received the faculty research grant from University Grants Commission, Nepal. The awarded fund goes to principal investigator official institute. Funders has no role in the study design, data collection, analysis, decision to publish, and preparation of manuscript.”

“There was no competing interest exists among authors.”

6. We note that you have indicated that data from this study are available upon request. PLOS only allows data to be available upon request if there are legal or ethical restrictions on sharing data publicly. For more information on unacceptable data access restrictions, please see http://journals.plos.org/plosone/s/data-availability#loc-unacceptable-data-access-restrictions.

7. We note that you have stated that you will provide repository information for your data at acceptance. Should your manuscript be accepted for publication, we will hold it until you provide the relevant accession numbers or DOIs necessary to access your data. If you wish to make changes to your Data Availability statement, please describe these changes in your cover letter and we will update your Data Availability statement to reflect the information you provide.

8. Please include your tables as part of your main manuscript and remove the individual files. Please note that supplementary tables (should remain/ be uploaded) as separate ""supporting information"" files.

Reviewers' comments:

Reviewer's Responses to Questions

**Comments to the Author**

1. Is the manuscript technically sound, and do the data support the conclusions?

Reviewer #1: Yes

Reviewer #2: Yes

Reviewer #3: Yes

2. Has the statistical analysis been performed appropriately and rigorously? 

Reviewer #1: I Don't Know

Reviewer #2: Yes

Reviewer #3: Yes

3. Have the authors made all data underlying the findings in their manuscript fully available?

Reviewer #1: Yes

Reviewer #2: Yes

Reviewer #3: Yes

4. Is the manuscript presented in an intelligible fashion and written in standard English?

Reviewer #1: Yes

Reviewer #2: No

Reviewer #3: Yes

5. Review Comments to the Author

Reviewer #1: - Change title:

The effect of educational intervention based on the theory of planned behavior in ….: Stochastic controlled trial

In the abstract, write the method of sampling

Write the mean value and standard deviation for all model constructs

Explain more in the conclusion

Match the keywords with the mesh in PubMed

Explain the reason and importance of the study in the introduction

In the introduction, write a review of the texts and a review of other studies in this field

Describe the application of the study in the introduction

Describe how to sample

Write the entry and exit criteria in the working method

Bring study restrictions

Write how to determine the sample size

Explain tools, is it self-made or standardized?

Write down the number of questions and the validity and reliability of the tool

Write how the tool scores

Write down the intervention and the strategies used

In the method, describe each learner for what type of analysis is used

In the results, write the standard deviation and mean values for all groups and variables

Write down the strengths and weaknesses of the study

Write study limitations

Bring suggestions

In the discussion section, compare the results with other studies and write your reasons

Reviewer #2: Dear editor in journal of Plos One

Thank you for your invitation to review of manuscript entitled “The effect of educational intervention on intention to perform breast self-examination and mammography among women of Pokhara Nepal: application of theory of planned behavior”

Introduction

Comment 1:

Introduction is too long. Please condense in to one half page.

Comment2:

Authors should be mention intervention studies based on TPB in introduction section.

Material and method

Comment 1:

Authors should be clarified setting samples were selected from a hospital, clinic or clinics.

Comment 2:

Authors should be clarified of how to educate illiterate people?

Comment 3:

Researchers need to figure out how to control confounders, especially in the control group.

Data collection instruments

Please mention numbers of items for constructs of TPB (attitude, subjective norms, perceived behavior control) and validity and reliability of its.

Discussion

Authors should be mention limitation of study.

Reviewer #3: This is an important and interesting a quasi-experimental controlled study to assess effect of an educational intervention on intention to perform breast examination and mammography among women of Pokhara Nepal. While this study has many merits such as using two intervention and control groups and the manuscript was written precise, it needs some revisions and English editing.

Below there are some comment about the manuscript and suggestions for its improvement:

Introduction

This test aids in the early detection of cases that are still curable.

This refers to BSE or CBE or mammogram?

Women should begin regular screening starting at age 45 years old. Women must have the chance to begin a yearly screening test between the ages of 40 and 44 years old.

Reference Please.

It focuses on women’s attitude and beliefs (21).

This sentence is not clear. Please revise the sentence.

A study done in Kathmandu Nepal, found out of 500 women 3.4%, 7.2% and 14.4% of women had undergone mammography, clinical breast exam and breast self-examination respectively.

I suggest write the abbreviation such as CBE…

If a woman believes she is at risk of breast cancer, she is more likely to undertake screening measures.

Reference please.

To date, there has been very little study on the effectiveness of educational intervention in determining women’s desire to go for breast cancer screening.

In Nepal or world? Reference please.

Study design and setting

A quasi-experimental controlled time series study with a control group design was conducted to assess the effectiveness of educational intervention on the intent to perform breast cancer screening among women in Pokhara, Nepal.

Please remove “with a control group”. It was repeted.

Respondents were selected by using purposive sampling. The sample size was determined using the Open Epi sample size calculator for a cross sectional study with a power of 80%, and a confidence level of 95%, with an assumption of study from mammography utilization (25).

Why “for a cross sectional study”??

Data collection instruments

The calculated sample size was 154 in the exposed group and 154 in the nonexposed group.

I suggest using consistent terminology throughout manuscript: interventional, exposed /intervention and non-exposed/control.

The questionnaire was developed by the researchers using the theory of planned behavior principles, considerable reviews, and expert comments.

I suggest using constructs instead of principles.

Outcome variables

The intention to go for monthly breast self-examination and mammography every two years was the primary result of this research.

There is secondary outcome as well?

The theory of planned construction of behavioral intention based on score cut-off points.

It is not clear.

Data collection

Every time the same tool was used to collect the information, the intervention group was given a repeated intervention in a time frame.

Please clarify how much time before using the tool occurred intervention.

Ethics statement

The comparison group provided the same informative session at the end of the research.

The comparison group was provided the same informative session at the end of the research.

At the conclusion of the research, the control group offered the identical instructional session.

The comparison group was provided the same informative session at the end of the research.

I think two sentences offer the same meaning! Please revise it.

Data analysis

Tables were used to represent the information.

Please remove it.

Results

I suggest removing the first sentence.

The average age in the interventional and control groups was 49.

I suggest writing about baseline features not the intervention and control groups.

… from 35.6 percent at baseline

35.6%.

High intention was found in subsequent posttest 1 (87.2%) and dropped in posttest

2 (37.2%) and again increased in posttest 3 (83.6%).

In which group?

Further, results showed that after intervention at 4 months and 12 months, significant differences were found between the intervention and control groups except in the 8 months regards referents

with peers.

It is not clear.

Then, there was no significant change observed even after the intervention was given in the control factors in the experimental groups.

Please revise it.

Generally, I suggest removing repeated sentences and reduce the results section.

Discussion

I suggest, in the first paragraph, report the main results of the study. And the explanation about TPB move to method or remove here.

I suggest removing repeated sentences and directly discussing the main points of the issue in the women. The discussion needs to be improve and reduce.

6. PLOS authors have the option to publish the peer review history of their article (what does this mean?). If published, this will include your full peer review and any attached files.

---

## [Author Response · Author response to Decision Letter 0]

10 Sep 2022

Reviewer 1: We have incorporated all the suggestions in revise manuscript. Thank you so much for the feedback, it was very helpful.

Reviewer 2: We thank you for the feedback. We addressed all the suggestions into revision. 

Reviewer 3: Thank you so much for the suggestions, it was very helpful, we incorporated all the feedback in the revision.

---

## [Editor Report · Decision Letter 1]

17 Oct 2022

PONE-D-22-05192R1The effect of educational intervention based on the theory of planned behavior on intention to perform breast self-examination and mammography among women of Pokhara, Nepal: A quasi-experimental studyPLOS ONE

Dear Dr. Rojana Dhakal,

Thank you for submitting your manuscript to PLOS ONE. After careful consideration, we feel that it has merit but does not fully meet PLOS ONE’s publication criteria as it currently stands. Therefore, we invite you to submit a revised version of the manuscript that addresses the points raised during the review process.

Please submit your revised manuscript by Dec 01 2022 11:59PM If you will need more time than this to complete your revisions, please reply to this message or contact the journal office at plosone@plos.org. Please include the following items when submitting your revised manuscript:A rebuttal letter that responds to each point raised by the academic editor and reviewer(s). You should upload this letter as a separate file labeled 'Response to Reviewers'.A marked-up copy of your manuscript that highlights changes made to the original version. You should upload this as a separate file labeled 'Revised Manuscript with Track Changes'.An unmarked version of your revised paper without tracked changes. You should upload this as a separate file labeled 'Manuscript'.If applicable, we recommend that you deposit your laboratory protocols in protocols.io to enhance the reproducibility of your results. Protocols.io assigns your protocol its own identifier (DOI) so that it can be cited independently in the future. For instructions see: https://journals.plos.org/plosone/s/submission-guidelines#loc-laboratory-protocols. Additionally, PLOS ONE offers an option for publishing peer-reviewed Lab Protocol articles, which describe protocols hosted on protocols.io. Read more information on sharing protocols at https://plos.org/protocols?utm_medium=editorial-email&utm_source=authorletters&utm_campaign=protocols.

We look forward to receiving your revised manuscript.

Kind regards,

Siew Chin Ong, PhD

Academic Editor

PLOS ONE

Journal Requirements:

Additional Editor Comments :

The authors have made a good attempt to respond to the issues raised by the editor and reviewers. However, Some of the concerns raised have not been sufficiently responded to. A convincing justification should be provided if the authors have a superior opinion to the comments raised.

To ensure the Editor and Reviewers will be able to recommend that your revised manuscript is acceptable, please pay careful attention to each of the comments that have been raised. This way we can avoid future rounds of clarifications and revisions, moving swiftly to a decision. Also, ensure that the revised manuscript is edited for LANGUAGE.

Examples:

Reviewer 1

"Change title: The effect of educational intervention based on the theory of planned behavior ....: Stochastic controlled trial."

"Theory of planned behavior" is not a MESH term."

"Separate sub-heading for inclusion and exclusion criteria."

"Indicate clearly all the relevant p-values in the description of respective tables."

Reviewer 3

"Overall English editing is needed?" e.g. "Introduction: "These" not "This"."
---

## [Author Response · Author response to Decision Letter 1]

10 Dec 2022

Dear Reviewers,

Thank you for the valuable feedback. We made all the corrections as recommended by the editors and reviewers. Please see the attached response to reviewers for more details.

---

## [Editor Report · Decision Letter 2]

18 Jan 2023

Attitude sustains longer than subjective norm and perceived behavioral control: Results of Breast Cancer Screening Educational Intervention

PONE-D-22-05192R2

Dear Dr Rojana,

We’re pleased to inform you that your manuscript has been judged scientifically suitable for publication and will be formally accepted for publication once it meets all outstanding technical requirements.

Kind regards,

Siew Chin Ong, PhD

Academic Editor

PLOS ONE
---

## [Editor Report · Acceptance letter]

30 Jan 2023

PONE-D-22-05192R2 

Attitude sustains longer than subjective norm and perceived behavioral control: Results of Breast Cancer Screening Educational Intervention 

Dear Dr. Dhakal:

I'm pleased to inform you that your manuscript has been deemed suitable for publication in PLOS ONE. Congratulations! Your manuscript is now with our production department. 

Kind regards, 

on behalf of

Dr. Siew Chin Ong 

Academic Editor

PLOS ONE